# Effect of Shading on the Morphological, Physiological, and Biochemical Characteristics as Well as the Transcriptome of Matcha Green Tea

**DOI:** 10.3390/ijms232214169

**Published:** 2022-11-16

**Authors:** Xi Chen, Kun Ye, Yan Xu, Yichen Zhao, Degang Zhao

**Affiliations:** 1The Key Laboratory of Plant Resource Conservation and Germplasm Innovation in Mountainous Region (Ministry of Education), College of Life Sciences/Institute of Agro-Bioengineering, Guizhou University, Guiyang 550025, China; 2Guizhou Plant Conservation Technology Application Engineering Research Center, Guizhou Institute of Prataculture/Guizhou Institute of Biotechnology/Guizhou Academy of Agricultural Sciences, Guiyang 550006, China; 3College of Tea, Guizhou University, Guiyang 550025, China

**Keywords:** tea, shade, chlorophyll, physiological, biochemical, transcriptome

## Abstract

High-quality tea leaves are required for matcha production. Shading is one of the key agronomic practices that can increase the quality of green tea. The objectives among matcha tea producers include increasing the ammonia and chlorophyll contents of tea buds, decreasing tea polyphenol contents, and enhancing tea aroma formation. In this study, Fuding white tea plants were cultivated under open-air conditions (control) as well as under 85% (S85) and 95% (S95) shade. The chlorophyll contents were highest for the S85 treatment, followed by the S95 and control treatments. Moreover, shading increased the theanine and caffeine contents, while decreasing the polyphenol (epicatechin and epigallocatechin) contents, thereby optimizing matcha tea flavors. A total of 2788 differentially expressed genes (DEGs) were identified, of which 1151 and 1637 were respectively upregulated and downregulated in response to shading. The GO and KEGG enrichment analyses indicated that most of the DEGs were associated with metabolic processes (e.g., MAPK signaling, plant-pathogen interactions, and phenylpropanoid biosynthesis). Therefore, shading may modulate tea plant metabolism, signaling, biosynthetic activities, and environment-related changes to gene transcription. The expression of amino acid permeases (APP) encoding genes was downregulated in tea plants. Thus, shading influences theanine biosynthesis and the AAP-mediated distribution of theanine in tea plants.

## 1. Introduction

Tea (*Camellia sinensis* L.), which is an economically valuable plant species, is used to produce popular beverages that are consumed worldwide because of their health benefits [1,2,3]. Tea quality is influenced by the presence of various metabolites, including amino acids (especially theanine), polyphenols, caffeine, and aroma compounds [4,5,6,7,8]. The consumption of traditionally brewed tea results in the intake of the water-soluble components of tea leaves. However, current tea-brewing practices, including the grinding of whole leaves to produce matcha (i.e., green tea) powder, enable the consumption of all of the nutrients in tea leaves. Matcha is popular because of its desirable fresh taste and color. Thus, there are various commercially available matcha-derived products. The matcha industry in China has recently developed rapidly, and the demand for matcha and matcha-derived products is increasing, especially Guizhou matcha, but the quality of the available products remains low, which has had detrimental effects on sales and exports. Matcha production is optimized by the use of high-quality green tea raw materials. More specifically, cultivating tea plants in the shade is critical for producing high-quality green tea. Ideally, individual tea plants or the entire tea garden should be shaded to decrease the radiation from the sun by 60–98% (typically by 85%). The changes to the morphological, physiological, and biochemical characteristics of tea plants induced by the cultivation under shade conditions include increases in the chlorophyll content [9]. Furthermore, earlier studies [9,10,11] demonstrated that shading leads to an increase in the amino acid content, but a decrease in the abundance of tea polyphenols (especially catechin), which are the main compounds responsible for the fragrance of green tea. The free amino acid and chlorophyll contents are crucial determinants of the quality and economic value of tea leaves. The market value of green tea is directly proportional to the amino acid content [12]. Therefore, the cultivation of tea plants under shade conditions has been promoted and applied to enhance the quality of tea leaves used in the matcha industry.

In this study, the open-air control samples were compared with those cultivated under 85% (S85) and 95% (S95) shade in terms of the morphological characteristics of tea buds, chlorophyll, and theanine contents, and gene expression to clarify the physiological, biochemical, and molecular mechanisms underlying the effects of shading on the quality of green tea. The results of this study provide tea researchers and growers with relevant information for increasing the quality of matcha or other green teas that are rolled during processing.

## 2. Results

### 2.1. Effect of Shading on Chlorophyll and Carotenoid Contents

Compared with the other treatments, the S85 treatment resulted in darker green leaves (Figure 1A–D) and significantly higher carotenoid (Figure 1I), total chlorophyll, chlorophyll a, and chlorophyll b contents as the duration of the shade treatment increased (Figure 1E,F). The rank-order for the chlorophyll a, chlorophyll b, and total chlorophyll contents among treatments was S85 > S95 > control (Figure 1E,F,H). On day 15 of the shade treatment, the chlorophyll b content for the S85 treatment was 1.07 mg g^−1^, which was 1.4-fold higher than that of the control treatment. Similarly, the chlorophyll b content for the S95 treatment was 1.01 mg g^−1^, which was 1.3-fold higher than that of the control treatment (i.e., significant increase) (Figure 1F). The rank-order for the carotenoid content among treatments was S85 > S95 > control. On day 20 of the shade treatment, the carotenoid content for the S85 treatment was 1.3-fold higher than that of the control treatment. The chlorophyll a/b ratio decreased as the shade treatment period increased. Moreover, the rank-order for the chlorophyll a/b ratio among the treatments was control > S85 > S95 (Figure 1G). Accordingly, shading increased the chlorophyll b content more than the chlorophyll a content (Figure 1).

### 2.2. Influence of Shading on Tea Morphology and Growth Parameters

Shading obviously affected the morphological characteristics of the young branches and leaves of the new tea plants. Under the S85 conditions, the leaf blade length increased during the treatment period; the increases were greater than those observed for the control and S95 treatments (Figure 2C). Leaf thickness decreased as the shade treatment time increased, indicating that shading may induce leaf thinning (Figure 2B). On day 15 of the shade treatment, the branch length was greater for the unshaded (control) plants than for the plants under the S85 and S95 conditions. On day 20 of the shade treatment, the branch length was significantly lower for the plants undergoing the S95 treatment than for the control plants or the plants maintained under S85 conditions, implying that the S85 treatment was better for promoting tea shoot growth than the other two treatments (Figure 2A). The analyses of tea leaf growth revealed that the fresh and dry weights decreased significantly as the shade treatment time increased. On day 20 of the shade treatment, the fresh weight for the control treatment was 70.3 mg, which was 1.15-times and 1.94-times greater than that of the S85 (60.86 mg) and S95 (36.15 mg) treatments, respectively these differences were significant. The leaf water contents for the S85 and S95 treatments were 79.55% and 79.80%, respectively, which were both significantly higher than the control leaf water content (75.78%). In contrast, there was no significant difference in the leaf water content between the S85 and S95 treatments (Figure 2D–F). On day 15 of the shade treatment, the number of new branches and dormant branches was higher for the control treatment (143) than for the S85 (121.33) and S95 (99.33) treatments (Figure 2G,H). The 100-bud weight decreased as the treatment time increased, with a rank-order of control > S95 > S85 (Figure 2I), indicating that moderate shading (i.e., S85) is conducive to the formation of tea branches.

### 2.3. Effect of Shading on the Amino Acid Content

The amino acid content of tea plants increased in response to shading. More specifically, the rank-order for the theanine content was S95 > S85 > control (Figure 3A). The rank-order for the abundance of glutamine, which is the precursor of theanine, was S85 > S95 > control (Figure 3C). Hence, light intensity is an important factor in promoting glutamine production, with excessive shading detrimental to the accumulation of glutamine. Shading also increased the serine and glutamate contents (Figure 3B,D). The arginine and aspartic acid contents increased during the S85 treatment period but decreased over time under the S95 conditions (Figure 3E,F). Thus, moderate shading appears to increase the amino acid content, whereas excessive shading has the opposite effect. Furthermore, in the cloudy Guizhou mountainous region (low latitude and high altitude), tea plants likely have a relatively high amino acid content, which is favorable for the production of matcha tea with a desirable taste.

### 2.4. Effect of Shading on Catechin and Caffeine Contents

The shading of tea plants decreased the epicatechin (Figure 4A), epicatechin gallate (Figure 4B), epigallocatechin (Figure 4C), and epigallocatechin gallate (Figure 4D) contents, whereas it increased the caffeine levels (Figure 4E).

### 2.5. Transcriptome Sequencing, Unigene Assembly, and Annotation

To analyze the molecular mechanism underlying the increase in the chlorophyll and theanine contents and the decrease in the tea polyphenol levels, the leaf transcriptomes of tea plants grown under control and S85 conditions were analyzed using the BGISEQ-500 platform, with three biological replicates per sample. An average of 6.30 Gb RNA-seq data were generated per sample and then evaluated in terms of quality. The RNA-seq data are summarized in Table 1. After the raw data were filtered to eliminate low-quality reads, 41,414,838 of the original 44,897,890 reads (i.e., 92.24%) were retained as clean reads. The clean reads were aligned to the reference genome sequence. The average read alignment rates for the genome and gene sets were 73.86% and 53.91%, respectively. A total of 46,032 expressed genes were detected, of which 36,250 were known genes. There were 9782 predicted new genes, with transcripts detected for these genes. There were also 31,221 new transcripts. Of the 21,061 novel alternative splice isoforms, 10,160 belonged to known protein-coding genes. Additionally, these transcripts belonged to novel protein-coding genes. A total of 45.28 × 10^6^ and 46.27 × 10^6^ raw reads were obtained for the control and S85 treatments, respectively. The filtering of the raw reads resulted in 42.01 × 10^6^ (Q20: 95.98%; Q30: 93.38%) and 42 × 10^6^ (Q20: 95.86%; Q30: 90.27%) clean reads for the control and S85 treatments, respectively. Of the clean reads for the control and S85 treatments, 74.04% and 73.68% were respectively mapped to reference sequences, including 36.88% and 36.61% that were uniquely mapped (Table 1). The results reflected the reliability of the transcriptome data, which were appropriate for the subsequent analysis.

On the basis of the method described by Wang et al. [13], the significant DEGs were identified using the following thresholds: at least a 2-fold difference in expression and a Q-value of 0.001. A total of 2788 DEGs were detected, of which 1151 were upregulated and 1637 were downregulated (Figure 5).

### 2.6. Gene Ontology (GO) and Kyoto Encyclopedia of Genes and Genomes (KEGG) Enrichment Analyses of the Differentially Expressed Genes

The GO terms in the three main categories (molecular function, cellular component, and biological process) were used to functionally characterize the significant DEGs. The highly enriched GO terms were cell process, metabolic process, cellular anatomical entity, intracellular, binding, and catalytic activity. A total of 1116 genes were annotated with catalytic activity, 988 genes were annotated with binding, 1169 genes were annotated with a cellular anatomical entity, and 666 genes were annotated with cell process and metabolic process (Figure 6A).

The KEGG metabolic pathways in the following seven branches were enriched among the DEGs: cellular processes, environmental information processing, genetic information processing, human disease (animals only), metabolism, organismal systems, and drug development. Further functional classifications were completed for each branch. The most common KEGG metabolic pathway branch among the DEGs was metabolism, followed by genetic information processing. The KEGG enrichment analysis revealed the DEGs were significantly associated with various pathways, including the MAPK signaling pathway, plant-pathogen interaction, and phenylpropanoid biosynthesis (Figure 6B). Accordingly, changes in metabolism and signal transduction as well as environment-related changes in gene expression levels may be induced by shade treatments. The functional annotation of the DEGs on the basis of GO and KEGG analyses provides researchers with important information relevant for future investigations on the effects of shade on tea plants.

In an earlier study on the metabolome of tea plants cultivated under shade, Dong et al. [14] observed that the theanine content differed significantly between the shaded and control plants, with the shade treatment increasing the abundance of theanine. AAPs are one of the two major families of amino acid transporters in plants. In tea plants, theanine is mainly synthesized from glutamate and ethylamine in the roots. It is transported to the shoots via AAPs and then hydrolyzed in the leaves [15,16,17]. A recent study indicated that the expression levels of theanine synthesis-related genes (e.g., GOGAT- and GS-encoding genes) and *CsAAP* genes in the roots are significantly upregulated in response to shade treatments, which may be associated with the increase in the leaf theanine content [17]. On the basis of the transcriptome sequencing data analyzed in the current study, we examined three AAP-encoding genes that were differentially expressed between the control and S85 treatment groups. The results indicated that the expression levels of these genes in tea leaves tended to decrease in response to shading (Figure 7A). Considered together, these findings suggest that shading affects theanine biosynthesis-related enzyme activities, while also potentially influencing the distribution of theanine among diverse tea plant tissues via AAPs.

Because theanine metabolism is closely related to nitrogen metabolism, the enzymes involved in nitrogen metabolism may also be important for theanine metabolism. Thus, we analyzed the expression profiles of genes involved in nitrogen metabolism, which revealed three DEGs encoding specific enzymes (i.e., carbonic anhydrase, glutamate synthase, and nitrate reductase) (Figure 7B) that mediate theanine biosynthesis. Therefore, we speculated that shade-induced changes to the theanine concentration may be related to the differential expression of these genes.

Pyruvate is an important intermediate for glucose metabolism as well as other metabolic pathways. The analysis of the transcriptome data resulted in the identification of DEGs encoding proteins in the pyruvate metabolic pathway, including key enzymes contributing to glycolysis (e.g., 6-phosphate fructokinase) and the conversion of pyruvate to other compounds (e.g., pyruvate decarboxylase) (Figure 7C), suggesting that shading regulates the changes in the expression of genes encoding enzymes in the glycolytic pathway.

### 2.7. Differential Gene Expression Levels Were Verified by Quantitative Real-Time Polymerase Chain Reaction (qRT-PCR)

To further validate the RNA-seq data, the expression levels of seven randomly selected genes that were differentially expressed between the S85-treated and control tea plants were analyzed in a qRT-PCR assay. The selected genes encoded the following enzymes: AAP (CSS000586, CSS0012351, CSS0013308), nitrate reductase (CSS0027199), and 6-phosphofructokinase 1 (CSS0022015). The gene expression trends revealed by qRT-PCR were consistent with the RNA-seq data (Figure 8), reflecting the reliability of the transcriptome sequencing analysis.

## 3. Discussion

Tea plants were originally grown in tropical forests, wherein the trees protected them from excessive light. Tea plants cultivated in tea gardens receive more sunlight than tea plants grown in forests. This increased exposure to light may lead to photoinduced stress and photoinhibition. There is evidence that intense solar radiation decreases photosynthetic activities in tea leaves, although the associated mechanism is unclear [18]. To enhance tea quality, most tea growers have adopted shading as part of their cultivation practices, while also decreasing energy consumption. Shading is one of the agronomic practices used to cultivate tea plants for the production of matcha or other green teas. Earlier studies confirmed that decreasing the light intensity during the growth of tea plants will affect the composition of quality-related components in tea buds, while also decreasing the polyphenol content in the whole plant, thereby enhancing the taste of green tea [19,20,21]. The catechin contents decreased in the leaves of tea plants grown under shade mesh. The four main catechins in the tea plants were (−)-epicatechin, (−)-epicatechin 3-gallate, (−)-epigallocatechin, and (−)-epigallocatechin gallate, of which (−)-epigallocatechin was the compound with the highest activity and content. Among teas, matcha is the best source of this catechin. Shading also increased the theanine and caffeine contents. Its unique chemical composition and desirable flavor distinguish matcha from other tea beverages. Moreover, matcha is considered to be the highest quality tea [3,9]. Most of the related research has focused on matcha processing and baking [22,23]. The present study was conducted primarily to clarify the effects of shading on tea plant morphological, physiological, and biochemical characteristics as well as the underlying molecular mechanisms.

In the current study, the shade treatment significantly affected several tea plant morphological parameters, including fresh and dry weights, the number of new and dormant branches, bud weight, water content, leaf length, leaf width, leaf length-to-width ratio, leaf area, leaf circumference (S85 > control > S95), and leaf thickness and leaf spacing (control > S95 > S85). Thus, suitable shading may induce the thinning of tea plants (i.e., altered leaf spacing), which is conducive to decreasing the energy required for matcha production and increasing the yield. Similarly, Sano et al. [9] reported that shading decreases the thickness of tea leaves and significantly modulates the leaf shape and color, which is critical for the production of high-quality green tea. We also determined that appropriate shading can decrease the leaf thickness, whereas excessive shading (e.g., S95) has the opposite effect.

Cultivating under shade conditions can significantly alter the phenotype of tea plants. For example, compared with open-air conditions, shading results in darker green leaves, which is likely related to an increase in the leaf chlorophyll content [9,24,25]. In the current study, the rank-order of the carotenoid, chlorophyll a, chlorophyll b, and total chlorophyll contents was S85 > S95 > control, implying that moderate shading is ideal for increasing the chlorophyll content of tea buds. Previous studies showed that shading can double the chlorophyll concentration of albino tea leaves, promote leaf regeneration, and increase carotenoid levels by 30%, with increases in the chlorophyll content primarily responsible for improved albino tea leaf characteristics [26,27]. These findings are consistent with the results of our study in which shading increased the chlorophyll content of tea plants.

Saijo et al. [28] and Wang et al. [20] demonstrated that shading can increase the amino acid content of tea plant shoots and inhibit the accumulation of polyphenols, such as catechins, which is conducive to improving the quality of green tea. Yu et al. [29] and Xu et al. [27] reported that shading treatments decrease the catechin content, but have the opposite effect on the caffeine content. In the current study, the theanine and caffeine contents increased in response to the shade treatment, whereas the polyphenol contents (i.e., epicatechin and epigallocatechin) decreased. Sano et al. [9] and Matsunaga et al. [11] determined that the shade-induced changes to catechin, caffeine, and amino acid contents are influenced by shade intensity and treatment duration. Moreover, shading increases the amino acid content, while decreasing the epigallocatechin contents. This suggests that the increased freshness of tea derived from shaded plants may be related to increases in the theanine content and decreases in the polyphenol contents. Decreases in the catechin content may weaken the bitterness of tea, whereas increases in the amino acid content can enhance the freshness of tea flavors and ultimately increase the quality of tea.

The GO and KEGG analyses revealed that most of the identified DEGs were associated with metabolism (e.g., MAPK signaling pathway, plant-pathogen interaction, and phenylpropanoid biosynthesis), indicating that shading may affect metabolism, signaling, biosynthesis, and environment-related gene expression levels. Theanine is mainly synthesized from glutamate and ethylamine in tea roots [30,31] and then transported to the shoots through AAPs, wherein they are hydrolyzed in leaves [13,32,33]. Yang et al. [16] recently detected the shade-induced downregulated expression of genes encoding AAPs in tea leaves, confirming that shading affects theanine biosynthesis and distribution among tea plant tissues. More specifically, shading promotes the biosynthesis and distribution of theanine in diverse tissues via its effects on the enzymes in the theanine biosynthesis pathway and the AAP family of theanine transporters. We identified three AAP-encoding genes that were differentially expressed between the control and S85-treated samples, with shade-induced downregulated expression trends in the leaves. These results indicate that shading modulates theanine biosynthetic enzymes and AAP-mediated theanine distribution in tea plants.

The expression profiles of the genes associated with nitrogen metabolism revealed genes encoding carbonic anhydrase, glutamate synthase, and nitrate reductase that were differentially expressed between the control and shaded tea plants. These enzymes have crucial functions related to theanine synthesis. Accordingly, the changes in the theanine content under shade conditions are likely related to the differential expression of these genes and genes encoding 6-phosphate fructokinase and pyruvate decarboxylase. Hence, shading appears to modify the production of enzymes that participate in the glycolytic pathway. Therefore, moderate shading is an agronomic practice that may enhance the production of high-quality green teas, including matcha.

This study exploited the adaptability of Fuding white tea and the unique climate conditions of Guizhou to elucidate the effects of different light intensities on tea plants, with implications for selecting ideal tea varieties and the optimal shading treatment for producing high-quality matcha tea. We also investigated the expression of genes involved in theanine synthesis and transport in tea plant roots, stems, and leaves. The generated data clarified how the expression of theanine synthesis- and transport-related genes is regulated by shading. The findings of this study may be useful for increasing the theanine content of matcha green tea.

## 4. Materials and Methods

### 4.1. Plant Materials and Treatments

Fuding white tea plants grown for 10 years in a tea garden (Guizhou Yuqing Fengxiangyuan Co., Ltd, Zunyi, China) were used as the test materials in this study. The test site (27°38′16″ N, 107°35′29″ E) located at an elevation of 914 m was a flat and trimmed part of the tea garden where all environmental conditions were the same, with the exception of the light conditions. The shade treatment was initiated on 3 April 2021, which is when most tea plants contained one bud and two leaves, and it ended approximately 3 weeks later on April 22. In addition to the control plants that were exposed to open-air conditions (i.e., no shade), other plants were maintained under 85% (S85) and 95% (S95) shade. Samples were collected every 5 days (i.e., days 0, 5, 10, 15, and 20). Three replicates were collected per time-point. The samples were immediately frozen in liquid nitrogen and then stored at −80 °C.

### 4.2. Analysis of Physiological and Biochemical Indicators

#### 4.2.1. Examination of Tea Morphology

A quadrat survey was conducted at 0, 5, 10, 15, and 20 days after shading. A square wire frame (400 cm^2^) that was 20 cm long on each side was used to randomly select three replicate areas comprising similarly growing tea plants. Leaves were collected from the plants and then immediately measured and weighed. New tea shoots and dormant shoots were counted for each quadrat survey sample. The fresh weight of all collected shoots was measured using a balance. The shoots were dried (vacuum-microwave finish drying and oven drying) before measuring their dry weight. The weight of 100 randomly selected tea buds was also determined (i.e., 100-bud weight). Twenty randomly selected shoots with one bud and two leaves were used to measure the branch shoot length using a ruler, whereas the leaf blade length was determined using a leaf area meter. Leaf thickness was measured using a Vernier caliper, and the number of leaves was recorded. The third leaf from the apex of each of the 20 shoots was harvested for the subsequent measurement of the leaf chlorophyll content. Other biochemical parameters were analyzed using shoots with one bud and two leaves that were ground to a powder.

#### 4.2.2. Determination of Free Amino Acid, Catechin, and Caffeine Contents

The free amino acid, catechin, and caffeine contents were determined according to the independent detection method of the Agricultural Biotechnology Platform Center, Institute of Biotechnology, Chinese Academy of Agricultural Sciences. Specifically, tea samples were ground to a powder using the MM 400 mixer mill (Retsch, Arzberg, Germany) (30 Hz, 1.5 min) and then lyophilized in the Christ ALPHA 1-2 LD plus freeze dryer for 24 h. After adding 100 mg lyophilized powder to a centrifuge tube, 1 mL 70% methanol was added and the solution was vortexed for 1 min. The sample was incubated at 70 °C in a water bath for 10 min (vortexed once after 5 min) and then cooled to room temperature. The solution was centrifuged at 21,428 rcf for 10 min, after which the supernatant was passed through a 0.22 µm filter membrane. The filtrate was added to a liquid phase inlet sample bottle to detect free amino acids and polyphenols. This analysis was performed using the 1290 ultra-high performance liquid chromatography system (Agilent Technologies, Palo Alto, CA, USA) and the 6420 mass spectrometer (for tandem mass spectrometry) (Agilent Technologies, Palo Alto, CA, USA). The contents of the following free amino acids were determined: theanine, glutamate, arginine, aspartic acid, glutamine, and serine. The contents of the following four catechins and caffeine were also determined: epicatechin, epicatechin gallate, epigallocatechin, and epigallocatechin gallate. The corresponding standards were purchased from Shanghai Jinye Biological Co., Ltd.

#### 4.2.3. Determination of the Chlorophyll Content

The second leaf of tea plants was wiped with clean gauze and cut into small pieces with scissors. A 0.1 g sample was mixed with 10 mL 95% ethanol (*v*/*v*) solution containing 5% acetone and then ground to a powder. After overnight incubation, the sample was shaken two or three times and then added to an ELISA plate. The absorbance of the solution (645 and 663 nm) was measured using the SpectraMax^®^ ABS Plus microplate reader (Molecular Devices, Sunnyvale, CA, USA). The chlorophyll content was calculated using the following formulae: chlorophyll a = 12.70A_663_ − 2.69A_645_; chlorophyll b = 22.9A_645_ − 4.68A_663_; A_645_ and A_663_ refer to the absorbance at 645 and 663 nm, respectively. The total chlorophyll content was determined as follows: chlorophyll a + chlorophyll b.

#### 4.2.4. RNA Extraction, cDNA Library Construction, and Sequencing

For the transcriptome sequencing (RNA-seq) analysis, cDNA libraries were constructed and sequenced by the Bgi Genomics Co., Ltd (Shenzhen, China). Total RNA was extracted from the tea leaves using CTAB (Ambion, Austin, TX, USA) and then the RNA concentration and integrity were determined using the RNA 6000 Nano Reagents Part 1 kit and the 2100 Bioanalyzer (Agilent Technologies, Palo Alto, CA, USA). The mRNA containing the poly-A tail was enriched using oligo(dT) magnetic beads. The rRNA was hybridized into a DNA probe, with the resulting DNA/RNA selectively digested first with RNaseH and then with DNaseI to eliminate the DNA probe. The desired RNA was obtained after a purification step. The obtained RNA was fragmented in a fragmentation buffer and then random hexamer primers were used for the reverse transcription to synthesize the first-strand cDNA sequence. After generating the second cDNA strand, the resulting double-stranded cDNA ends were modified to produce a blunt and phosphorylated 5′ end and a 3′ sticky end with a protruding A-tail. A bubble-like junction with a protruding T-tail was ligated to the 3′ ends. The ligation products were amplified by PCR using specific primers. The PCR products were thermally denatured into single-stranded DNA and then a bridge primer was used to obtain a single-stranded circular DNA library, which was sequenced using the BGISEQ-500 sequencing platform.

#### 4.2.5. Unigene Assembly and Annotation

High-quality clean reads were obtained by filtering out the raw reads to eliminate low-quality reads, adapter contamination, and reads with an unknown base (N) content greater than 5%. The retained clean reads were saved in the FASTQ format. Clean reads were aligned to the reference genome sequence using HISAT software. Briefly, HISAT anchors part of each sequence on the genome according to the global FM index. It then compares the remaining sequence of each read with the local index of the alignment positions to extend the aligned region [34]. The clean reads were aligned to reference gene sequences using Bowtie2, whereas the gene expression levels were calculated using RSEM [35,36].

The clean reads were then aligned separately to the reference genome using the HISAT software, thereby enabling the quantitative analysis of gene/transcript expression in each sample. The overall transcriptome sequencing analysis included an examination of the randomness of the transcripts, the transcript coverage and distribution, and sequencing saturation. The reference transcriptome sequence lengths were analyzed using a Perl script. For each sample, transcripts were reconstructed using StringTie and then the reconstructed transcripts of all samples were integrated using Cuffmerge. The integrated transcripts were compared with the reference annotation information using Cuffcompare, and transcripts with class code types ‘u’, ‘i’, ‘o’, and ‘j’ were defined as new transcripts. The new transcripts that were predicted to encode proteins by CPC were added to the reference gene sequences to complete the reference sequence information for the subsequent analysis. The unigene open reading frames (ORFs) were detected using getorf, whereas the ORFs were aligned to the transcription factor protein domain (data from TF) using hmmsearch. The unigenes were identified according to the transcription factor family features described in the plant transcription factor database PlantTFDB. The aligned genes were annotated using the PRGdb plant resistance gene database and the DIAMOND software. The annotation results were further screened on the basis of the alignment coverage (query coverage) and consistency (identity) to identify possible disease-resistant genes in plants [37,38,39].

#### 4.2.6. Gene Expression Analysis and Detection of Differentially Expressed Genes

After aligning the clean reads to the genome sequence and calculating the gene expression levels for each sample [35,36], the Pearson correlation coefficient was calculated using the cor function in the R software. A principal component analysis was performed using the princomp function in the R software; the results were visualized using the ggplot2 package in the R software. The DEGseq method, which is based on a Poisson distribution model, was used to detect DEGs as previously described by Wang et al. [13]. A 2-fold or more difference in expression and a Q-value of 0.001 were set as the criteria for identifying significant DEGs. A hierarchical cluster analysis of the DEGs was performed using the heatmap function in the R software. The enrichment analysis of the DEGs was performed using the phyper function in the R software.

#### 4.2.7. Gene Ontology and Kyoto Encyclopedia of Genes and Genomes Enrichment Analyses

The GO terms are divided into the following three functional categories: molecular function, cellular component, and biological process. The DEGs were functionally characterized on the basis of the significantly enriched GO terms (FDR-corrected *p*-value and Q-value ≤ 0.05), which were determined using the phyper function in the R software. Additionally, the significantly enriched KEGG pathways (FDR-corrected *p*-value and Q-value ≤ 0.05) among the DEGs were determined using the phyper function in the R software.

#### 4.2.8. Quantitative Real-Time Polymerase Chain Reaction Analysis

To verify the accuracy and reproducibility of the RNA-seq data, a qRT-PCR analysis was performed using gene-specific primers that were designed using the Primer 6.0 software (Table 2). The PrimeScript™ RT kit (TaKaRa Biotechnology Co., Ltd., Dalian, China) was used to convert the extracted RNA (1 µg) into first-strand cDNA. The qRT-PCR analysis was performed using NovoStart^®^ SYBR qPCR SuperMix Plus (Novoprotein, Suzhou, China) and the CFX Connect Real-Time PCR system (Bio-Rad, Hercules, CA, USA). The analysis was completed using three biological replicates and three technical replicates. The *CsGAPDH* gene was selected as the internal reference control. Gene expression levels were calculated according to the 2^−ΔΔCt^ method.

## Figures and Tables

**Figure 1 ijms-23-14169-f001:**
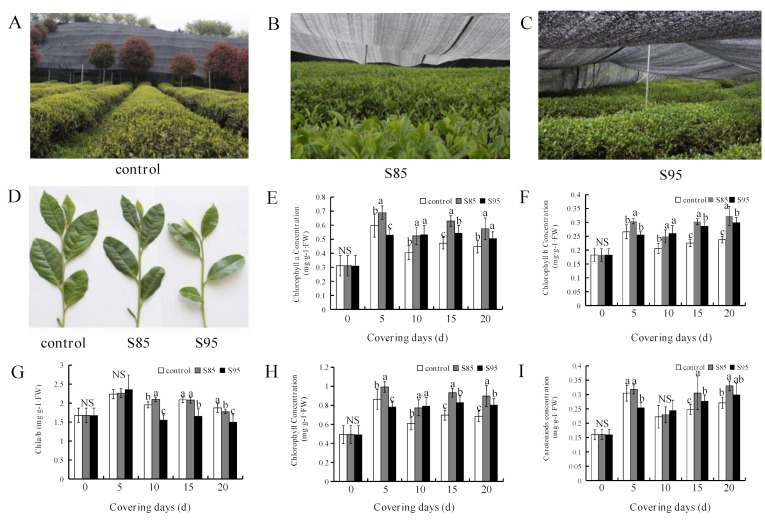
Effects of shading on the appearance as well as the chlorophyll and carotenoid contents of tea plants. (**A**) Open-air (control) tea plants (**B**) Tea plants grown under 85% shade (**C**) Tea plants grown under 95% shade (**D**) Appearance of tea plant leaves after 20 days of growth (**E**) chlorophyll a content (**F**) Chlorophyll b content (**G**) Chlorophyll a/b ratio (**H**) Total chlorophyll content (**I**) Carotenoid content. ‘NS’ and different letters above the bar indicate insignificant and significant (*p* < 0.01; n = 3 biological replicates) differences, respectively, between the control and shaded plants.

**Figure 2 ijms-23-14169-f002:**
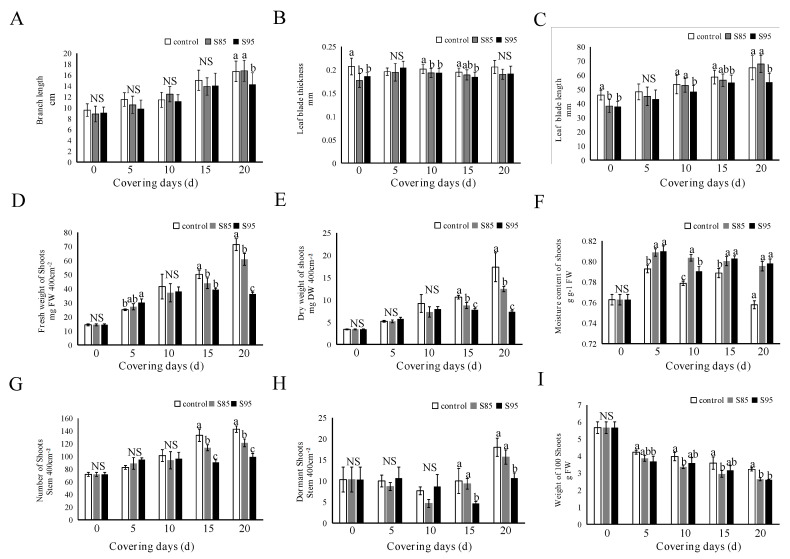
Effects of shading on tea leaf morphology and traits. (**A**) Branch length; **(B**) Leaf thickness; (**C**) Leaf length; (**D**) Fresh weight; (**E**) Dry weight; (**F**) Water content; (**G**) Number of branches (400 cm^2^); (**H**) Number of dormant branches (400 cm^2^); (**I**) 100-bud weight. ‘NS’ and different letters above the bar indicate insignificant and significant (*p* < 0.01; n = 3 biological replicates) differences, respectively, between the control and shaded plants.

**Figure 3 ijms-23-14169-f003:**
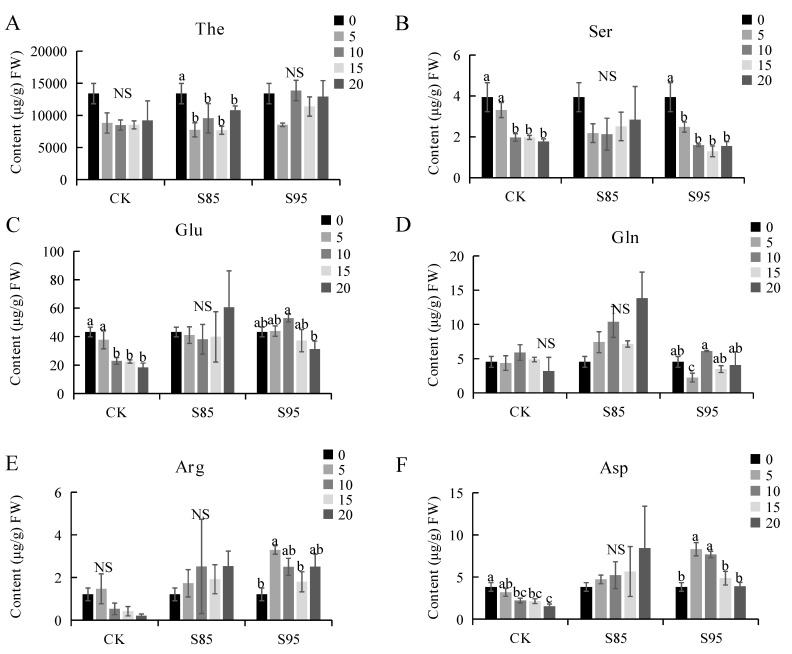
Effect of shading on the amino acid content of tea leaves. (**A**) Theanine content; (**B**) Serine content; (**C**) Glutamate content; (**D**) Glutamine content; (**E**) Arginine content; (**F**) Aspartic acid content. CK is the control. ‘NS’ and different letters above the bar indicate significant (*p* < 0.01; n = 3 biological replicates) differences between the control and shaded plants.

**Figure 4 ijms-23-14169-f004:**
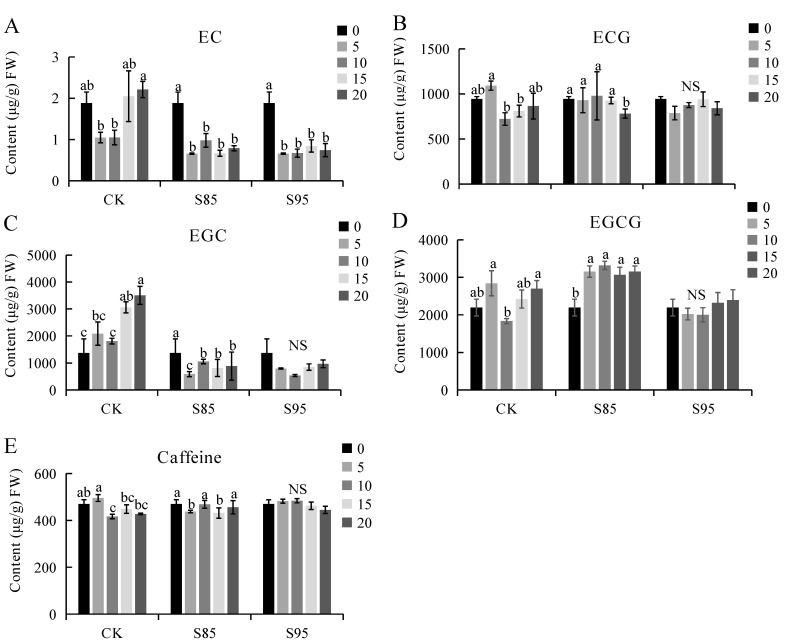
Effect of shading on the catechin and caffeine contents in tea leaves. (**A**) Epicatechin (EC); (**B**) Eepicatechin gallate (ECG); (**C**) Epigallocatechin (EGC); (**D**) Epigallocatechin gallate (EGCG); (**E**) Caffeine. CK is the control. ‘NS’ and different letters above the bar indicate significant (*p* < 0.01; n = 3 biological replicates) differences between the control and shaded plants.

**Figure 5 ijms-23-14169-f005:**
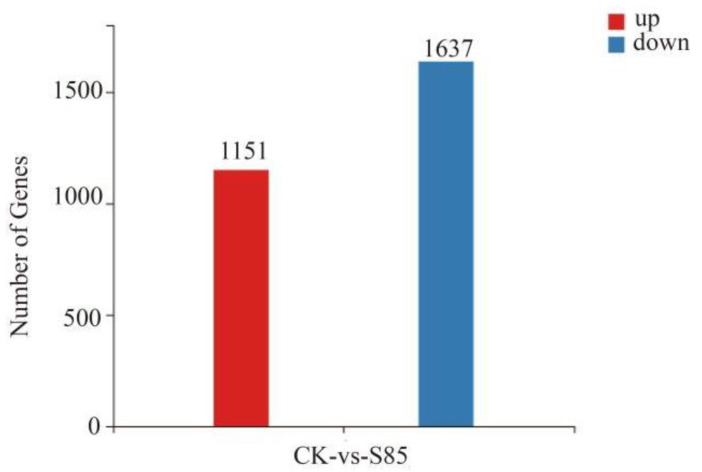
Number of differentially expressed genes between the control and S85 treatment groups.

**Figure 6 ijms-23-14169-f006:**
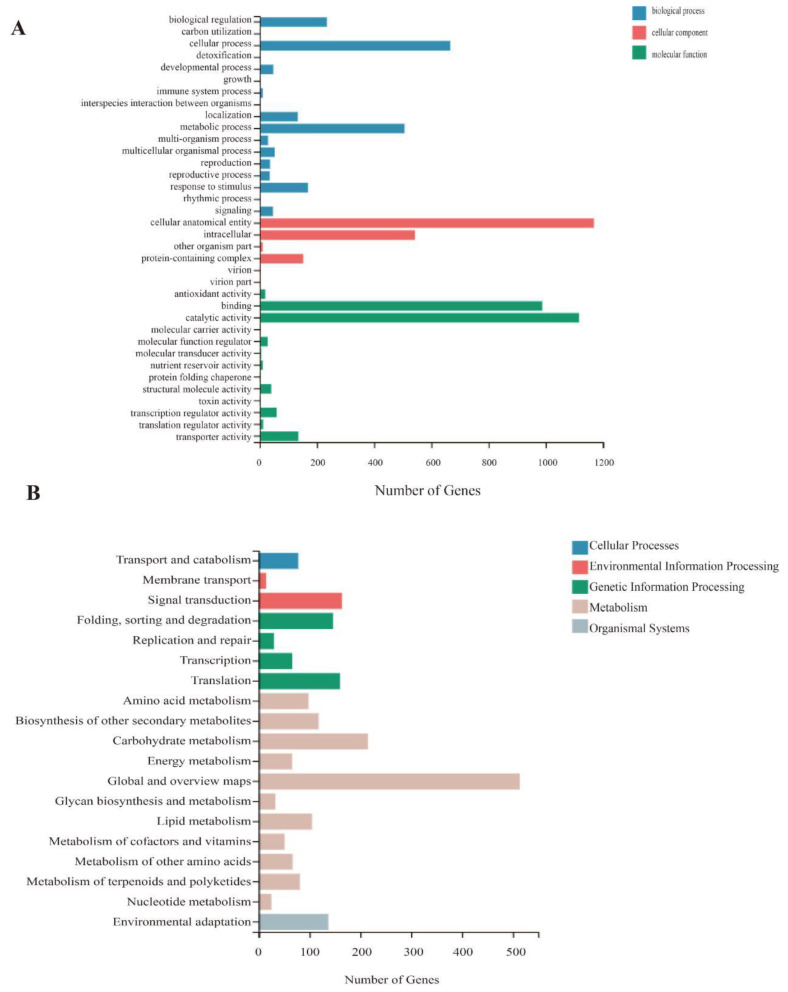
GO and KEGG classification of differentially expressed genes between the control and S85 treatment groups. (**A**) The x-axis presents the number of genes annotated with a particular GO term, whereas the y-axis presents the GO terms used for the functional classification (**B**) The x-axis presents the number of genes assigned to a particular KEGG pathway, whereas the y-axis presents the enriched KEGG pathways.

**Figure 7 ijms-23-14169-f007:**
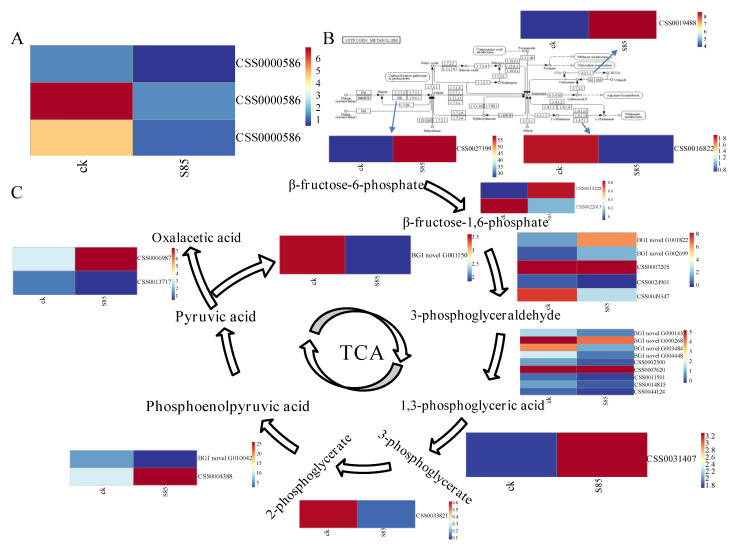
Differential expression profiles. (**A**) Genes related to pyruvate metabolism (**B**) Genes involved in nitrogen metabolic pathways (**C**) Amino acid permease genes.

**Figure 8 ijms-23-14169-f008:**
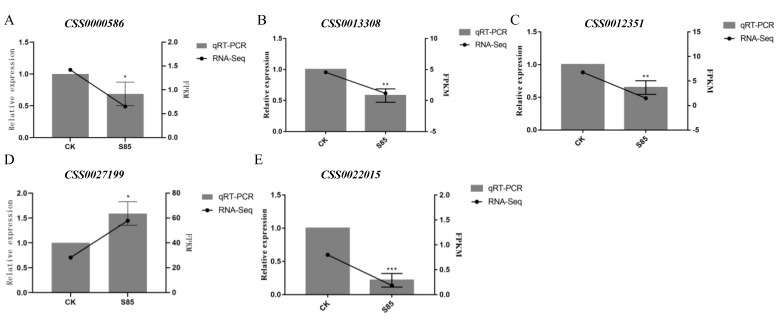
Validation of the differentially expressed genes by a qRT-PCR analysis. Significant differences between CK and S85 are indicated (* *p* < 0.05, ** *p* < 0.01 and *** *p* < 0.001).

**Table 1 ijms-23-14169-t001:** Overview of the RNA-seq data.

Sample	Raw Reads	Clean Reads	Q20/%	Q30/%	Clean Reads Ratio/%	Total Map/%	Unique Map/%
Control	45.28 × 10^6^	42.01 × 10^6^	95.98	93.38	92.78	74.04	36.88
S85	46.27 × 10^6^	42 × 10^6^	95.86	90.27	90.82	73.68	36.61

**Table 2 ijms-23-14169-t002:** Primers for the qRT-PCR analysis of differentially expressed genes.

Primer Name	Primer Sequences (5′-3′)	Sequence Length
CSS0000586-F	AGCAGTGATTGGGTCAGGAG	123 bp
CSS0000586-R	GCGAGGAGAGTAGAAGTGTAGT	
CSS0012351-F	CGCCTGAAGAGAACTGGAAC	177 bp
CSS0012351-R	CTGTGAGAGGAGATTGGAAGTG	
CSS0013308-F	CGCCTGAAGAGAACTGGAACT	180 bp
CSS0013308-R	GCACTGTGAGAGGAGATTGGAA	
CSS0027199-F	ACCCTTCACAAACACCACCTC	167 bp
CSS0027199-R	CGGTTCCAGCGTTGATGAGA	
CSS0022015-F	GACCCAAGCCAAATACTTCGT	112 bp
CSS0022015-R	GGCATAGCAATCCACCTTCTT	

## Data Availability

Not applicable.

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
