# Peer review of "Effect of Shading on the Morphological, Physiological, and Biochemical Characteristics as Well as the Transcriptome of Matcha Green Tea"

_ijms, 2022, doi:10.3390/ijms232214169_

Round 1
Reviewer 1 Report
The study deal with an interesting concern: the effects of different light intensities on tea plants, with implications for selecting ideal tea varieties and the optimal shading treatment for producing high-quality matcha tea.
The subject is relevant and the work is on a good level. The manuscript fits within the scope of the journal, is clearly written and contains extensive collection of experimental studies.
Reviewer 2 Report
The study is interesting, with practical and future economic significance.
Remarks:
Figure 1E-I and Figure 2 - need to describe the measurement of the abscissa.
Figure 2 - cm2, as elsewhere, the square should be superscript formatted.
The description of the abbreviation AAP appears in the third citation of the term - line 200.
The acronym in Figure 3, on the abscissa CK is probably the control, but it needs to be clarified in the text under the figure.
A significant omission is a need for statistical processing of the results of tea morphology, free amino acid content, catechin, caffeine, and chlorophyll content. Statistical analysis should be applied based on which the authors can claim that there is a difference between the analyzed samples in the parameters they are investigating!
The discussion should be more detailed!
Round 2
Reviewer 2 Report
I want to thank the authors for considering my comments.